# Social networks predict gut microbiome composition in wild baboons

Jenny Tung[1,2,3,4]*, Luis B Barreiro[5], Michael B Burns[6,7], Jean-Christophe Grenier[5], Josh Lynch[6,7], Laura E Grieneisen[8], Jeanne Altmann[3,9], Susan C Alberts[2,3], Ran Blekhman[6,7], Elizabeth A Archie[3,8]*

[1]Department of Evolutionary Anthropology, Duke University, Durham, United States; [2]Department of Biology, Duke University, Durham, United States; [3]Institute of Primate Research, National Museums of Kenya, Nairobi, Kenya; [4]Duke Population Research Institute, Duke University, Durham, United States; [5]Department of Pediatrics, Sainte-Justine Hospital Research Centre, University of Montreal, Montreal, Canada; [6]Department of Genetics, Cell Biology, and Development, University of Minnesota, Minneapolis, United States; [7]Department of Ecology, Evolution, and Behavior, University of Minnesota, Minneapolis, United States; [8]Department of Biological Sciences, University of Notre Dame, Notre Dame, United States; [9]Department of Ecology and Evolutionary Biology, Princeton University, Princeton, United States

**Abstract** Social relationships have profound effects on health in humans and other primates, but the mechanisms that explain this relationship are not well understood. Using shotgun metagenomic data from wild baboons, we found that social group membership and social network relationships predicted both the taxonomic structure of the gut microbiome and the structure of genes encoded by gut microbial species. Rates of interaction directly explained variation in the gut microbiome, even after controlling for diet, kinship, and shared environments. They therefore strongly implicate direct physical contact among social partners in the transmission of gut microbial species. We identified 51 socially structured taxa, which were significantly enriched for anaerobic and non-spore-forming lifestyles. Our results argue that social interactions are an important determinant of gut microbiome composition in natural animal populations—a relationship with important ramifications for understanding how social relationships influence health, as well as the evolution of group living.

*For correspondence: jt5@duke.edu (JT); earchie@nd.edu (EAA)

**Competing interests:** The authors declare that no competing interests exist.

## Introduction

Vertebrate intestines are home to thousands of bacterial species that exert profound effects on their hosts: they train the immune system, produce vitamins, help resist pathogens, and contribute substantially to daily energy acquisition (*Bergman, 1990*; *Turnbaugh et al., 2006*; *Hooper et al., 2012*; *Bengmark, 2013*; *Morgan et al., 2013*). In humans, inter-individual variation in gut microbiome composition has repeatedly been linked to major health concerns, including obesity, diabetes, cancer, heart disease, and autoimmune disorders (e.g., *Turnbaugh et al., 2009*; *Hooper et al., 2012*; *Bengmark, 2013*; *Iida et al., 2013*; *Koeth et al., 2013*; *Viaud et al., 2013*).

However, despite its importance, large gaps remain in our understanding of the forces that shape gut microbiome composition. Among the least understood but potentially most significant such forces are the effects of host social interactions. From an evolutionary perspective, social effects on the gut microbiome may be an underappreciated consequence of group living, associated with both

**eLife digest** The digestive system is home to a complex community of microbes—known as the gut microbiome—that contributes to our health and wellbeing by digesting food, producing essential vitamins, and preventing the growth of harmful bacteria. The recent development of rapid genome sequencing techniques has made it much easier to identify the species of microbes found in the gut microbiome, and how this microbiome's composition varies between individuals.

Studies in humans and other primates suggest that direct contact during social interactions may alter the composition of the gut microbiome in an individual. This could explain why there is a strong association between social interactions and health in humans and other social animals. However, similarities in the gut microbiomes of individuals within a social group could also be due to a shared diet or a common environment. The information collected during long-term studies of wild primates offers an opportunity to analyze and assess the influence of diet, environment and social interaction on the gut microbiome.

Here, Tung et al. studied the gut microbiomes of 48 wild baboons belonging to two different social groups in Amboseli, Kenya. Using a technique called shotgun metagenomic sequencing, they sequenced DNA extracted from samples of feces collected from individual baboons. The sequence data revealed that an individual's social group and social network can predict the species found in its gut microbiome. This remained the case even when other factors—such as diet, kinship, and shared environments—were taken into account.

Tung et al.'s findings suggest that direct physical contact during social interactions may be important in transmitting gut microbiomes between members of the same social group. However, scientists still don't know whether this exchange is good or bad for the health of the baboons. Future work will try to understand whether baboons benefit from acquiring gut microbes from their group members, and if the gut microbes of some social groups are better than others.

fitness costs and benefits (*Lombardo, 2008*; *Archie and Theis, 2011*; *Ezenwa et al., 2012*; *Montiel-Castro et al., 2013*). For example, co-housing in lab mice promotes the transmission of bacterial communities that contribute to inflammatory bowel disease, implicating social relationships in microbiome-associated disease risk (*Garrett et al., 2010*). In bumblebees, socially transmitted gut bacteria protect against a widespread and virulent gut parasite, suggesting that socially mediated microbial transmission can also confer powerful benefits (*Koch and Schmid-Hempel, 2011*). If social interactions predict gut microbiome composition in free-living vertebrates as well, this link could help explain the strong association between social interactions and health in highly social species (e.g., *Berkman and Syme, 1979*; *House et al., 1988*; *Sapolsky, 2004*; *Holt-Lunstad et al., 2010*).

A handful of recent studies in humans and other primates provide circumstantial evidence for social effects on the gut microbiome (*Degnan et al., 2012*; *Kinross and Nicholson, 2012*; *Yatsunenko et al., 2012*; *Song et al., 2013*). For instance, in wild chimpanzees, social group membership predicts the identity and abundance of gut microbes, while kinship, age, and sex do not (*Degnan et al., 2012*). In humans, shared residence predicts gut microbiome similarity (*Kinross and Nicholson, 2012*; *Yatsunenko et al., 2012*; *Song et al., 2013*). To date, these effects have largely been attributed to shared diets, as members of the same household or social group tend to consume similar foods in similar proportions (*Claesson et al., 2011*; *Kinross and Nicholson, 2012*; *Yatsunenko et al., 2012*). However, social relationships could also shape gut microbiomes more directly, via transmission from shared environments (*Lax et al., 2014*) or during physical contact.

Differentiating between these mechanisms requires fine-grained data on social interactions, shared environments, and diet. Such complementary data sets are rare, but are frequently collected in long-term primate field studies. Here, we leveraged one such study, on the intensively studied Amboseli baboons of Kenya (*Alberts and Altmann, 2012*), to test whether social group structure and social interactions within groups predict either the taxonomic or the functional composition of the gut microbiome. Like humans, baboons are highly social, group-living primates. Members of the same social group travel together, consume similar foods, and drink from the same water sources. Within social groups, individuals selectively engage in frequent affiliative grooming interactions, which solidify social bonds and have the potential to mediate bacterial transmission. Within this context, we

addressed three central questions. We first asked (i) does social group membership predict gut microbiome composition, as shown for humans and chimpanzees (*Degnan et al., 2012*; *Kinross and Nicholson, 2012*; *Yatsunenko et al., 2012*; *Song et al., 2013*)? We then asked two novel questions that have not been addressed in prior studies: (ii) within social groups, do rates of social interactions (captured here by grooming-based social networks) predict gut microbiome similarity after accounting for dietary patterns, shared environments, and kinship? And (iii) which bacterial species, with what lifestyle traits, are most likely to be socially transmitted, both between and within social groups?

## Results

We generated shotgun metagenomic data for the distal gut using fecal samples from 48 members of two baboon social groups ('Mica's group' or 'Viola's group'; one sample per individual; *Supplementary file 1*). Together, these individuals represented almost complete sampling (92%) of the adult members of both groups. Fecal samples were collected during a single 1-month timespan to minimize developmental, temporal, and seasonal heterogeneity. During this time, Mica's group and Viola's group exploited adjacent home ranges, with the center of each range separated by just a few kilometers (*Figure 1A*).

Using the program MetaPhlAn 2.0 (*Segata et al., 2012*), we identified 925 bacterial and archaeal taxa to the species level and quantified their relative abundance ($364 \pm 150$ s.d. species per sample; *Figure 1—figure supplement 1*; *Supplementary files 2, 3*; see *Supplementary files 3, 4* for parallel results using a de novo assembly approach). We also identified and quantified the relative abundance of 9013 microbial-encoded enzyme orthologs using the HUMAnN pipeline (mean $\pm$ SD = $2746 \pm 560$ KEGG orthologs per sample; *Abubucker et al., 2012*; *Figure 1—figure supplement 3*; *Supplementary file 5*). The taxa we found comprised a typical primate gut microbiota, dominated by the phyla Firmicutes (mean $\pm$ SD = $42.2\% \pm 8.4\%$), Proteobacteria ($13.0\% \pm 2.8\%$), Actinobacteria ($9.4\% \pm 4.6\%$), and Bacteroidetes ($7.3\% \pm 2.4\%$) (*Figure 1—figure supplement 1, 5*; *Supplementary file 6*). In some samples, especially those in Viola's group, we also detected a large contribution from the phylum Spirochaetes, consistent with findings from other primates, ancient humans, and modern-day human hunter-gatherers (*De Filippo et al., 2010*; *Tito et al., 2012*; *Schnorr et al., 2014*).

### Social group membership was the strongest single predictor of gut microbiome composition

Across all 48 individuals, social group membership explained 18.6% of global variation in gut microbial species composition (as summarized by a Bray–Curtis dissimilarity matrix; PERMANOVA for social group effects: $p < 10^{-4}$; *Figure 1C*). Social group membership was also the dominant source of variance in the abundance of enzyme gene orthologs encoded by gut microbes, explaining 10.8% of global variance in a Bray–Curtis dissimilarity matrix (PERMANOVA: $p = 0.003$; *Figure 1D*). In contrast, sex, age, and sequencing read depth made comparatively minor or non-significant contributions to gut microbiome composition (PERMANOVA: sex, age and read depth explained 3.6%, $p = 0.026$; 5.3%, $p = 0.052$; 6.0%, $p = 0.024$ of variance in taxonomic composition, respectively; no significant variation was explained by sex, age, or read depth for enzyme gene orthologs). Furthermore, social group remained a strong and significant predictor of taxonomic and enzyme gene ortholog composition even after controlling for genetic relatedness between study subjects (partial Mantel test for taxonomic composition: $r = 0.378$, $p < 10^{-5}$; for enzyme gene orthologs: $r = 0.140$, $p = 1.6 \times 10^{-3}$).

### Differences in gut microbiome composition between social groups were unlikely to be explained by diet

Previous associations between social proximity and gut microbial composition in humans and other primates have largely been attributed to diet (*Degnan et al., 2012*; *Kinross and Nicholson, 2012*; *Yatsunenko et al., 2012*). However, the two social groups in our study inhabited a relatively homogeneous savannah environment and exploited very similar resources. During the sample collection period, half of each group's diet was devoted to grass corms, and similar proportions were devoted to other food types, including grass seed heads, *Acacia tortilis* seed pods, leaves (primarily grass blades), and *Acacia xanthophloea* gum (*Figure 1B*; *Supplementary file 7*). The only diet component that differed significantly between the two groups was the proportion devoted to fruit

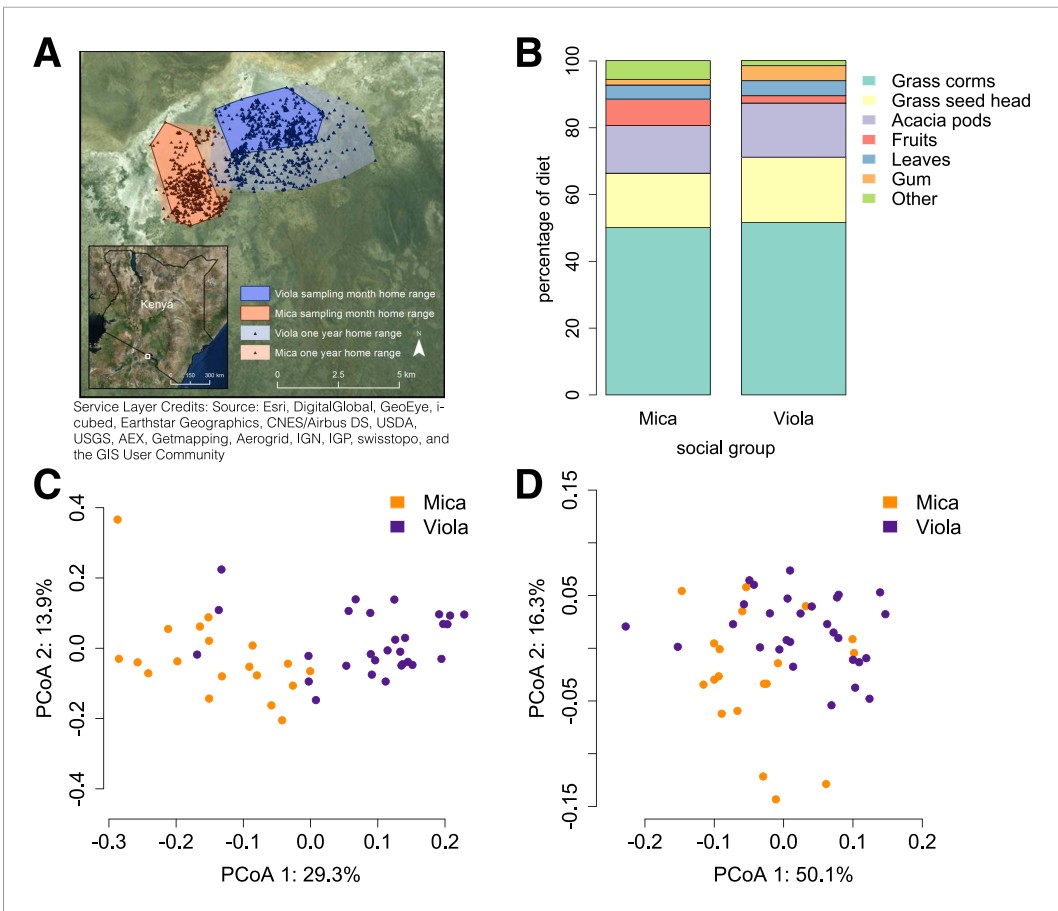

**Figure 1**. Social group membership predicts microbiome composition. (**A**) Group home ranges in the year prior to and during sample collection. (**B**) Diet composition during sample collection. Only the proportion of fruit consumed significantly differed between groups (p = 0.05; *Supplementary file 7*). Principal coordinates plots of Bray–Curtis dissimilarity matrices for (**C**) taxonomic (*Supplementary file 2*) and (**D**) KEGG enzyme ortholog composition of individual gut microbiomes (*Supplementary file 5*). Social group membership explained significant variation in gut microbial composition (PERMANOVA: $r^2$ = 0.186, p < $10^{-4}$) as well as gut microbial enzyme ortholog composition ($r^2$ = 0.108, p = 0.003). Relative abundances of common bacterial phyla and KEGG enzyme orthologs are shown in *Figure 1—figure supplement 1, 2*. A rarefaction analysis of species-level sampling is shown in *Figure 1—figure supplement 3*. The results of the HUMAnN pipeline are shown in *Figure 1—figure supplement 4*. A comparison between baboon and human microbiome composition across body sites is shown in *Figure 1—figure supplement 5*.

The following figure supplements are available for figure 1:

**Figure supplement 1**. Proportional representation of common phyla in each sample.

**Figure supplement 2**. Proportional representation of common KEGG orthologs in each sample, summarized as pathways.

**Figure supplement 3**. Rarefaction analyzes of shotgun metagenomic data.

**Figure supplement 4**. HUMAnN pipeline results.

**Figure supplement 5**. PCA projection of baboon gut microbiome data and Human Microbiome Project data collected from different body sites.

(permutation test: p = 0.05). However, we found no differences between the two groups in the abundance of two common fruit-associated bacterial enzymes, pectinesterase (p-value for social group in a linear mixed effects model: p = 0.306) and pectate lyase (p-value for social group in a linear mixed effects model: p = 0.869). Furthermore, patterns of differential taxonomic abundance between groups did not recapitulate differences associated with differential consumption of fresh fruits and vegetables described in a human gut microbiome data set (*Davenport et al., 2014*; see 'Materials and methods').

## Grooming networks predicted gut microbiome composition within groups

Despite few detectable differences in diet, unidentified environmental differences between Mica's group and Viola's group could explain the differences in gut microbiome composition we observed. To test whether social contacts per se predicted gut microbiome composition, we turned to fine-grained data on within-group grooming interactions. Grooming is by far the most common form of physical contact in baboons. Importantly, the strength of grooming relationships between pairs of individuals in the same social group varies considerably, despite the fact that all members of a social group travel together and use the same resource base.

To test whether physical contact predicted gut microbiome composition, we constructed grooming networks for each social group, using all grooming interactions observed in the year prior to and during microbiome sampling (*Figure 2A,B*). We found that, in both groups, closer grooming partners harbored more similar communities of gut bacteria (Mantel test between Bray–Curtis microbiome dissimilarity matrices and social network matrices: Mica's group $r = -0.257$, $p = 3.0 \times 10^{-4}$; Viola's group $r = -0.173$, $p = 8.0 \times 10^{-4}$; *Figure 2C,D*). This pattern was not driven by host genetic effects: although female relatives have stronger grooming bonds, controlling for pairwise relatedness still produced strong support for a relationship between grooming and taxonomic composition for Viola's group (partial Mantel test controlling for kinship: $r = -0.148$, $p = 2.0 \times 10^{-3}$), and a consistent trend in Mica's group (partial Mantel test controlling for kinship: $r = -0.163$, $p = 0.060$). Interestingly, extending this analysis to the level of enzyme gene orthologs suggested that close grooming partners also have functionally more similar gut microbiomes. Grooming networks predicted variation in within-group enzyme gene ortholog abundance for Mica's group (partial Mantel test controlling for kinship: $r = -0.22$, $p = 0.014$), but not Viola's group (partial Mantel test controlling for kinship: $r = -0.051$, $p = 0.166$).

Despite the relative homogeneity of diet within social groups, our results could still be explained by a diet-related mechanism if close grooming partners consumed more similar diets. Alternatively, close social partners might experience similar environmental exposures if they used more similar microenvironments in the group's home range. We tested these possibilities directly, focusing on adult females for whom diet composition and spatial proximity data were routinely collected (N = 11 females in Mica's group and N = 20 females in Viola's group). Grooming network proximity also predicted microbiota composition in this restricted data set (Mantel tests: Mica's group: $r = -0.328$, $p = 9.0 \times 10^{-3}$; Viola's group: $r = -0.228$, $p = 2.6 \times 10^{-3}$), and remained a significant predictor of microbiota composition after accounting for dietary similarity (partial Mantel test controlling for dietary similarity: Mica's group $p = 0.020$; Viola's group: $p = 0.005$) and spatial proximity (partial Mantel test controlling for spatial proximity for Mica's group $p = 0.039$; Viola's group: $p = 0.005$). Additionally, we found no evidence that close social partners consumed more similar diets (Mantel tests: Mica's group: Mantel $r = -0.200$, $p = 0.080$; Viola's group: Mantel $r = 0.0942$, $p = 0.876$).

## Socially structured bacteria tended to be anaerobic and non-spore-forming

We next investigated which bacterial species were associated with the strong signature of social structure in our data set. To identify these 'socially structured' species, we focused on the 327 most prevalent species in our data set (i.e., those found in ≥50% of samples). Using a mixed effects model controlling for age, sex, read depth, and host genetic relatedness, we identified 64 species (19.6%, using a 10% false discovery rate) that were significantly differentially abundant in the two social groups. We performed a complementary analysis, using a test of spatial autocorrelation, to investigate whether close grooming partners exhibited similar bacterial abundances within social

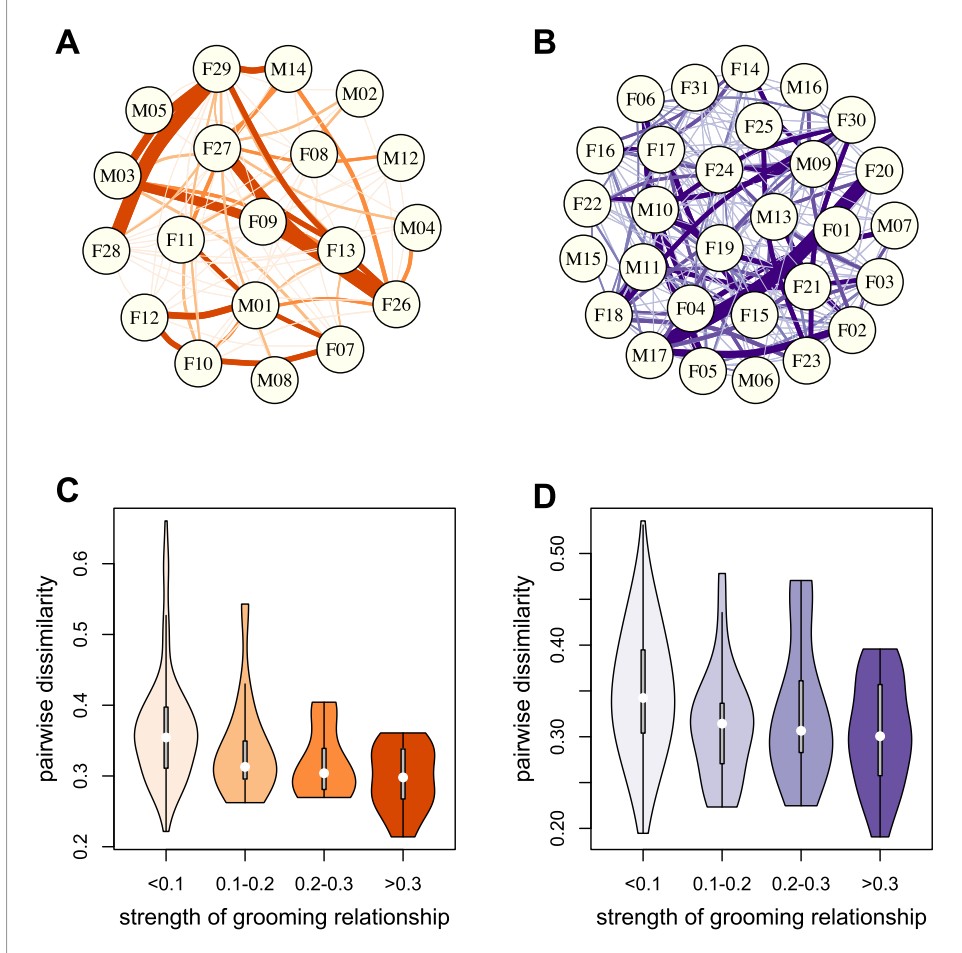

**Figure 2**. Grooming-based social networks predict microbiome composition. Social networks based on grooming interactions in the year prior to and including the month of microbiome sampling in (**A**) Mica's and (**B**) Viola's social groups. Each circle represents an individual (with the individual's ID listed within the circle). Lines represent grooming interactions between individuals, and heavier lines reflect stronger grooming relationships. (**C** and **D**) Violin plots depicting the relationships between pairwise grooming bond strength vs pairwise Bray–Curtis dissimilarity in taxonomic composition in Mica's and Viola's groups, respectively. White dots represent median values and grey rectangles represent the first and third quartiles of the data. Rotated kernel density plots representing the underlying data are shown on each side. Stronger bonds predict more similar gut microbiotas in both groups (Mica's group: Mantel test $r = -0.257$, p = $3.0 \times 10^{-4}$; Viola's group: $r = -0.173$, p = $8.0 \times 10^{-4}$). Parallel results based on de novo assembly are shown in *Figure 2—figure supplement 1*.

The following figure supplement is available for figure 2:

**Figure supplement 1**. Evidence for social structuring of the gut microbiome based on de novo assembly.

groups as well (due to the larger sample size, we performed these tests in Viola's group; see 'Materials and methods'). Among the same set of 327 prevalent species, we found 51 species (15.6%, 10% false discovery rate) for which proximity within the group's grooming network significantly predicted abundance (*Supplementary file 8*). Interestingly, 15 species were significantly socially structured both between groups and within social networks—more species than expected by chance (hypergeometric test, p = 0.020).

We next conducted an enrichment analysis to test whether the set of significantly socially structured species contained some taxonomic groups more often than by chance. We found that socially structured species were phylogenetically non-random at both between-group and within

social network levels of analysis (*Figure 3A,B*). Moreover, at both levels of analysis, similar taxonomic groups were significantly enriched for socially structured species (red asterisks on *Figure 3*), including the phylum Actinobacteria; the families Bifidobacteriaceae, Coriobacteriaceae, and Veillonellaceae; and the genus *Bifidobacterium*, a group of Gram-negative bacteria that has been linked to beneficial health effects in humans (*Servin, 2004*; *Gronlund et al., 2007*; *Turroni et al., 2008*). The striking similarities between the two levels of analysis suggest that common underlying mechanisms—mediated by direct social contact rather than diet or general physical proximity—account for both between-group differences and grooming network effects within groups.

Finally, we extended our enrichment analysis to test whether the set of socially structured species was enriched for particular bacterial lifestyles. We reasoned that, if socially structured species depend on direct transmission between baboons, as our data suggest, they should be less likely than other species to persist outside of a host. Thus, we predicted that socially structured species would tend to be anaerobic and unable to produce spores. To test these predictions, we turned to information about bacterial lifestyles available in the Genomes OnLine Database (*Pagani et al., 2012*), using both species-level (n = 138) and genus-level (n = 299) traits (see 'Materials and methods' for trait assignment criteria). We found that socially structured species were consistently enriched (relative to all species or genera tested) for an anaerobic, non-spore forming lifestyle (*Figure 3C*; hypergeometric tests for socially structured species between groups, species level traits: p = 0.017; socially structured species within group, species level traits: p = 0.067; socially structured species between groups, genus level traits: p = 0.036; socially structured species within group, genus level traits: p = 0.040). For

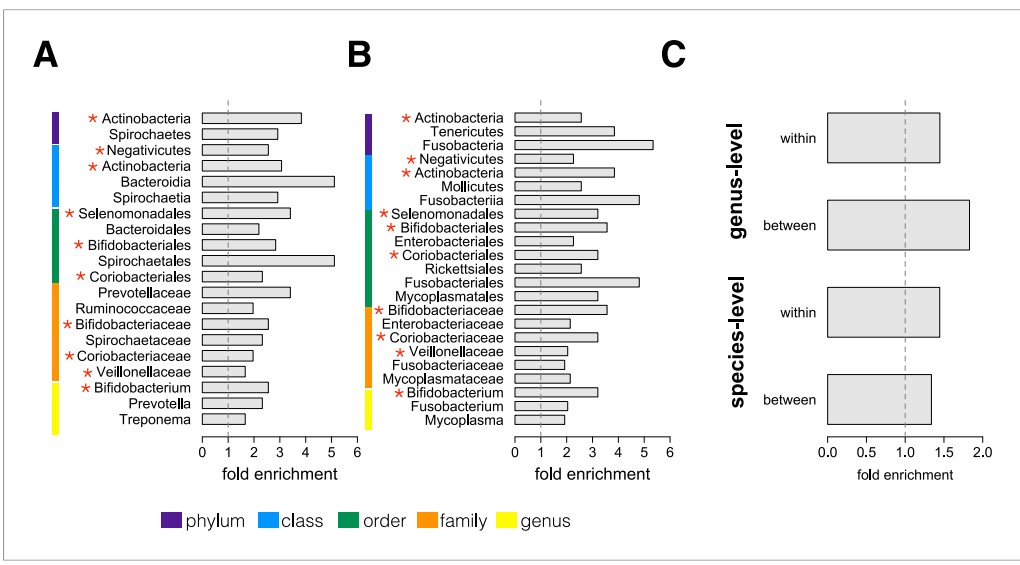

**Figure 3**. Socially structured species are taxonomically and phenotypically nonrandom. Bacterial taxonomic groups significantly enriched (10% FDR) for socially structured species (**A**) between social groups and (**B**) within the grooming network for Viola's group (*Supplementary file 8*). Vertical dashed lines depict a fold enrichment of 1, representing the background level of taxon abundance in the data set. Red asterisks denote taxonomic groups identified as significantly enriched at both levels of analysis. (**C**) Significant enrichment of anaerobic, non-spore-forming bacterial taxa, both between and within groups, at both species and genus levels (socially structured species between groups, species level traits: p = 0.017; socially structured species within group, species level traits: p = 0.067; socially structured species between groups, genus level traits: p = 0.036; socially structured species within group, genus level traits: p = 0.040). See *Figure 3—figure supplements 1, 2* for a comparison of the enrichment of p-values in our data set vs an empirical null distribution.

The following figure supplements are available for figure 3:

**Figure supplement 1**. Enrichment of low p-values in the data vs an empirical null: between group analyses.

**Figure supplement 2**. Enrichment of low p-values in the data vs an empirical null: within group network analysis.

instance, 17% of the species in this analysis differed significantly in abundance between social groups; however 32% of anaerobic and non-spore forming species were significantly socially structured. Notably, no species that were both aerobic and spore-forming were socially structured at the level of social groups or social networks, except for one case in the genus-level analysis.

## Discussion

Taken together, our results provide strong evidence that social interactions directly affect the composition of the gut microbiome in wild baboons. To our knowledge, this study is the first to test whether rates of interaction within cohabiting groups, as opposed to between groups or households, explain variation in the gut microbiome. Specifically, we found that an individual's contacts in a grooming-based social network, as well as its membership in a given social group, were highly predictive of its gut microbiome composition at both the species and genic levels. Unlike prior studies, we were able to exclude kinship, shared diet, and shared environment as the basis for our observations. Our results are thus unique among studies to date in the degree to which they implicate direct, affiliative physical contact as a determinant of gut microbiome composition in natural populations. Our data also provide the first evidence in vertebrates that social effects on the microbiome extend to its functional composition. These findings lend important support to the hypothesis that social interactions play a role in the health-related consequences of variation in gut microbiome composition (e.g., *Turnbaugh et al., 2009*; *Hooper et al., 2012*; *Bengmark, 2013*; *Iida et al., 2013*; *Koeth et al., 2013*; *Viaud et al., 2013*), with potentially important consequences for the evolution of sociality (*Lombardo, 2008*; *Archie and Theis, 2011*; *Ezenwa et al., 2012*; *Montiel-Castro et al., 2013*).

Thus, our results highlight the importance of socially mediated transmission in shaping gut microbiomes. However, unlike some prior studies in mice and bumblebees (*Garrett et al., 2010*; *Koch and Schmid-Hempel, 2011*), baboons are not coprophagic, raising a question about the mechanisms that facilitate gut microbial transfer between social partners. One possibility is that the duration and intimacy of grooming bouts, which include frequent hand-to-mouth contact, may be important in exposing baboons to the gut bacteria of their grooming partners. Furthermore, some grooming bouts, especially those directed from adult males to estrous females, concentrate heavily on the ano-genital region, increasing the probability of fecal-oral transfer. Such close contact may be especially important in the transmission of anaerobic, non-spore-forming species, as these bacteria are not thought to persist for long periods of time outside of a host (*Wilson, 2008*). However, some relatively hardy bacterial species may also be transmitted via social contact (*VanderWaal et al., 2013*), and recent modeling efforts suggest that fecal-oral transmission can be highly efficient in socially structured host populations, even when transmission is indirectly mediated through the soil (*Nunn et al., 2011*).

Interestingly, our observations suggest that social partners not only share more similar sets of gut microbes, but also similar abundances of individual microbial species. One explanation for this pattern is that when bacteria from a host colonize a social partner, they arrive pre-adapted to occupy the available gut microbial niches in their new host (*Walter and Ley, 2011*). Specifically, because members of a single bacterial species can have markedly different gene contents, a given member of a gut microbial species may perform different functions in different hosts (*Walter and Ley, 2011*; *Costello et al., 2012*). However, if social partners transmit bacteria with similar capabilities to each other, these bacteria may serve similar functions in both hosts and thus be found in similar abundances. This hypothesis could be further tested by assessing if bacterial species isolated from social partners tend to represent shared strains that perform similar biological functions.

In humans, affiliative physical contact (e.g., hugging, kissing, holding hands) is common and may provide a similar route through which close social partners transmit gut bacteria. In addition, surfaces in human homes may act as reservoirs for household-specific bacterial communities (*Lax et al., 2014*), possibly facilitating social transmission through intermediate surfaces. Future work, in both humans and animals, will be important to establishing the relative importance and generality of socially mediated transmission. In particular, population genetic studies have the potential to directly map the genetic structure of microbiome-associated species onto the social structure of host populations to test whether close social partners tend to share genetically more similar bacterial populations than non-partners (e.g., *VanderWaal et al., 2013*). Fine-grained studies of how gut microbial communities change in social species, before and after perturbations to their social networks, will also be important

for understanding the time scales on which social transmission of microbes act. Such efforts would also contribute an important longitudinal perspective. Our power to identify associations between social relationships and microbiome composition in this study was probably facilitated by our sampling scheme, which eliminated the contribution of temporal or seasonal effects. More comprehensive long-term studies will be valuable for placing these effects in context, alongside concomitant changes in season, diet, and resource use.

In humans, variation in the taxonomic and genic composition of the microbiome is increasingly linked to health issues, such as obesity and autoimmune disorders (e.g., *Turnbaugh et al., 2009*; *Hooper et al., 2012*; *Bengmark, 2013*; *Koeth et al., 2013*). Health and survival in social species (including humans and baboons) are also strongly associated with social relationships (*Berkman and Syme, 1979*; *House et al., 1988*; e.g., *Sapolsky, 2004*; *Silk et al., 2009*; *Holt-Lunstad et al., 2010*; *Silk et al., 2010*; *Archie et al., 2014*). However, few studies have connected these two observations. By highlighting the strong relationship between microbiome composition and social networks, our findings indicate the importance of further research in this area. One of the most important unanswered questions is whether social network-mediated microbiome sharing produces net fitness benefits or costs for hosts. Previous research on fecal-oral or social network-mediated transmission has focused almost exclusively on pathogens or parasites. Microbiome studies have the potential to broaden this perspective to include species with beneficial effects. Indeed, while we found several socially structured taxa that have been associated with pathogenic effects (e.g., *Fusobacterium* spp, *Campylobacter ureolyticus*), we found several other bacteria thought to be beneficial to hosts. For example, members of the phylum Actinobacteria, especially the genus *Bifidobacterium*, are commonly thought to have probiotic effects in humans due to their role in complex carbohydrate digestion, pathogen inhibition, and vitamin production (*Servin, 2004*; *Gronlund et al., 2007*; *Turroni et al., 2008*). Understanding the balance between social transmission of pathogenic vs commensal or beneficial bacteria thus promises to provide valuable new insight into the link between disease risk and the evolution of sociality.

## Materials and methods

### Study subjects, sample collection, DNA extraction, and metagenomic data generation

Study subjects were 48 wild, adult baboons living in the Amboseli ecosystem, a semi-arid savannah in southern Kenya (*Supplementary file 1*). The baboons were studied as part of the Amboseli Baboon Research Project (ABRP), which has been collecting continuous, individual-based data on all the members of several baboon social groups since 1971 (*Alberts and Altmann, 2012*). The specific subjects for this project represented near complete sampling (92%) of all the adult members of two social groups, called 'Mica's group' and 'Viola's group'. The baboons are individually recognized by experienced observers, who visit each group several times per week, year round, for 5-hr monitoring visits (*Alberts and Altmann, 2012*).

Distal gut microbiome composition was characterized using fecal samples collected opportunistically from known individuals. All fecal samples were collected during a single 1-month span in the dry season (7 July 2012 to 8 Aug 2012: *Supplementary file 1*). Samples were collected within a few minutes of defecation, thoroughly mixed, and then preserved in 95% ethanol (2:5 feces to ethanol). DNA was extracted from each sample using MO BIO's PowerSoil DNA Isolation kit, according to the manufacturer's instructions (MO BIO Laboratories, Inc., Carlsbad, CA). For each individual, 200 ng of extracted DNA were prepared for metagenomic sequencing on an Illumina HiSeq 2500 using the Kapa Biosystems Library Preparation Kits (Kapa Biosystems, Wilmington, MA). Specifically, DNA samples were sheared to an average size of 400 base pairs, ligated to barcoded adapters, and subjected to 100 base pair paired end sequencing at the UCLA Neuroscience Genomics Core. In total, we generated 1.4 billion raw, paired-end Illumina sequences across all samples (mean ± SD = 14.4 ± 13.7 million read pairs per sample). All raw reads are deposited in the National Center for Biotechnology Information (NCBI) Short Read Archive (BioProject PRJNA271618).

### Assessment of microbiome taxonomic composition using MetaPhlAn 2.0

Species-level taxonomic abundances were inferred for all samples using MetaPhlAn 2.0 (*Segata et al., 2012*). MetaPhlAn 2.0 estimates the relative abundance of bacterial species by mapping reads against

a set of clade-specific marker sequences, which unequivocally identify microbial clades at the species level or higher taxonomic levels. Based on 12,926 complete bacterial genomes, MetaPhlAn 2.0 is able to provide clade-specific markers for a total of 3848 bacterial species, 925 of which were detected in our data set (*Supplementary files 2, 3*). Specifically, we mapped our sequence reads against the clade-specific markers using the 'very-sensitive-local' alignment mode implemented in Bowtie 2 (*Langmead et al., 2009*). This mode produces alignments that can be trimmed at one or both extremes in order to optimize the alignment score. Because spurious or poor-quality reads are unlikely to match any of the pre-defined marker sequences, no preprocessing of the metagenomic DNA sequences was performed, as recommended by the authors. However, we tested the robustness of these estimates by re-running MetaPhlAn 2.0 on a subset of our data after trimming the reads to eliminate adapter sequences and bases with a quality score <20. Correlations between the bacterial abundance estimates obtained using unprocessed data and those obtained using the trimmed data were always above 0.97, confirming that MetaPhlAn 2.0 is indeed highly robust to potential sequence artifacts.

## Assessment of enzyme gene family composition

To investigate variation in the genic composition of the gut microbiome, we combined information from the Kyoto Encyclopedia of Genes and Genomes database (KEGG: *Kanehisa and Goto, 2000*; *Kanehisa et al., 2014*) with the HMP Unified Metabolic Analysis Network (HUMAnN) v0.99 pipeline (*Abubucker et al., 2012*). We first filtered the forward reads for quality using USEARCH v7.0 (*Edgar, 2010*). Specifically, for each sample, we (i) trimmed reads to a length of 99 bases, (ii) excluded reads shorter than 99 bases, and (iii) excluded reads with expected error (a measure of read quality in USEARCH based on base call quality and read length) > 0.5.

An average of 87.9% of all reads passed quality filtering (*Figure 1—figure supplement 4*). Remaining reads were translated in all three possible reading frames and aligned against a reduced KEGG database (last free version, June 2011) using the *ublast* function of USEARCH v7.0 and default parameters. The reduced KEGG database was generated by removing entries for which no KEGG orthology (KO) assignments existed and subsequently clustering each KO individually (*uclust* v1.5.579, using 85% sequence identity as the clustering cutoff) (*Edgar, 2010*; *Kanehisa and Goto, 2000*; *Kanehisa et al., 2014*). This database was converted to a USEARCH-compatible database file prior to running *ublast*. An average of 23.0% of the input reads across all samples were assigned an identity from the KEGG database (*Figure 1—figure supplement 4*). Finally, the *ublast* output was used as input for HUMAnN. HUMAnN was configured to generate KO abundances from BLAST hits of enzymes as well as coverage and abundances for KEGG pathways and modules.

## Social group membership and microbiome composition

To investigate the correlation between social group membership and the composition of baboon gut microbiomes, we constructed separate summaries of the complete taxonomic composition data set from MetaPhlAn 2.0 and the complete enzyme gene ortholog abundance data set from HUMAnN. Specifically, for each data set, we used the *vegdist* function in the R package *vegan* (*Oksanen et al., 2013*) to calculate a 48 × 48 Bray–Curtis dissimilarity matrix, which describes the global dissimilarity in gut microbial composition between each pair of individuals in the data set. To understand sources of variance in these matrices, we performed PERMANOVA analyses (*adonis* function in *vegan*) with 10,000 permutations. In addition to social group, predictor variables in this analysis were age, sex, and total read depth. Sex was known from direct observation of the study subjects. Ages were known to within a few days' error for 39 of the 48 individuals in the data set. The remaining 9 individuals immigrated into the population after birth, and so their ages were estimated using well-defined metrics and comparison to known-age animals (*Alberts and Altmann, 1995*). Of these 9 individuals, 6 animals had birth dates estimated to be accurate within 1 year, and 3 animals had birth dates estimated to be accurate within 2 years. All study subjects were adults (i.e., all females had attained menarche, and all males had attained adult dominance rank; *Onyango et al., 2013*).

To assess the possible confounding effects of kinship, we constructed a matrix of pairwise genetic relatedness values from the extensive pedigree data available for the Amboseli population (e.g., *Buchan et al., 2003*; *Alberts et al., 2006*; current pedigree includes 1409 individuals, with 1298 known maternal links and 526 known paternal links) using the R package *pedantics* (*Morrissey and*

*Wilson, 2010*). We then used partial Mantel tests to assess the correlation between a matrix describing group co-residency (cells took a value of 1 if two individuals resided in different groups and a value of 0 if they were co-resident) and the Bray–Curtis dissimilarity matrix for taxonomic composition, controlling for the pairwise genetic relatedness matrix.

## Differences in diet between social groups

To assess differences in diet between the two social groups, we used direct observations of the food consumed by adult female baboons in each group during the month in which samples were collected. Diet composition data were collected in the context of random-order focal animal sampling (*Altmann, 1974*). Specifically, ABRP observers spent 4 hr of each group visit rotating through the group, conducting focal animal samples on adult females in the order dictated by a randomized list. Each focal animal sample was 10 min long, during which activity (feeding, walking, resting etc) was recorded during point samples collected at 1-min intervals. When feeding was observed (353 point samples in Mica's group and 731 point samples in Viola's group), the observers recorded the type of food consumed. Food types were divided into 7 categories, including: (1) corms of all grass species, (2) seed heads of all grass species, (3) pods from *A. tortilis* and *A. xanthophloea* (4) fruits, including those from *Azima tetracantha*, *Salvadora persica*, *Solanum dubium*, *Trianthema ceratosepala*, and *Tribulus terrestris*, (5) leaves from *Lyceum* sp. and all grass species, (6) gum from *A. xanthophloea*, and (7) unknown/unidentified diet items (*Supplementary file 7*).

To calculate the contribution (including confidence intervals) of each of the seven major food categories to each group's diet, we conducted 1000 random subsamples of one foraging point sample per focal animal sample. We took this approach to avoid autocorrelation between point samples collected during the same 10-min focal sample. To test for differences in diet between groups, we repeated the same analysis after randomly permuting group membership across the females in our data set. We calculated the proportion of cases in which between-group differences in the proportion of a food consumed exceeded between-group differences in the 1000 permuted data sets. This proportion is equivalent to the p-value for the null hypothesis that the two groups did not differ in diet.

Because we detected a nominally significant difference ($p = 0.05$) in the amount of fruit consumed by the members of Mica's group (7.9%, 95% CI: 0.0–8.3%) and the members of Viola's group (2.2%, 95% CI: 1.0–7.3%) during the sampling period, we also compared our results to a published data set of seasonal differences in gut microbiome composition in humans (*Davenport et al., 2014*). These differences are believed to be the result of differences in consumption of fresh fruits and vegetables. Only three genera were detected as both significantly differentially abundant in the diet-related human data set (FDR = 10%) and significantly enriched for differential abundance between social groups in our data set, at a conservative (for comparative purposes) threshold of $p \leq 0.05$. *Bifidobacterium* was more abundant in humans when they consumed less fresh fruit; *Prevotella* and *Treponema* were more abundant when they consumed more fresh fruit. In our data set, however, *Bifidobacterium* levels were more abundant in Mica's group, which consumed more fruit, and *Prevotella* and *Treponema* species were more abundant in Viola's group, which consumed less fruit, suggesting that the patterns we observed are due to other sources of variance.

## Social interactions and microbiome composition within groups

To test whether grooming-based social networks predicted gut microbiome similarity, we constructed grooming networks based on *ad lib* observations of grooming interactions collected in the year prior to and including the period of microbiome sampling (8 August 2011 to 8 August 2012: 1648 total interactions, with 667 in Mica's group and 981 in Viola's group). *Ad lib* grooming interactions were collected throughout the monitoring visit while observers were carrying out focal animal sampling. *Ad lib* grooming data were used to calculate a count of observed grooming interactions between all adult dyads present in each social group (range = 0 to 41 interactions per dyad). These data were used to construct a matrix of grooming relationship strength by scoring the strongest dyadic grooming relationship in each group as a 1 and weighting all other dyadic relationships relative to this strongest bond. We then used Mantel tests to investigate the strength of the correlation between group-specific grooming networks and group-specific Bray–Curtis dissimilarity matrices, constructed as described above. We used partial Mantel tests to assess whether grooming network-microbiome dissimilarity matrix correlations were driven by kinship (represented using pedigree-based pairwise relatedness estimates).

To investigate alternative explanations for social network effects on the microbiome, we collated data on diet and spatial proximity for members of each social group, focusing on adult females only (comparable data were not available for adult males). Parallel to the time span for social network construction, we compiled data for the year prior to and including the month of microbiome sampling. For diet, we extracted all foraging-related point samples from the females in our microbiome data set (1380 points in Mica's group; 1989 points in Viola's group). We subsampled each data set so that only one point sample was represented per focal sample, which avoids autocorrelation between point samples collected during the same focal. We then constructed a table of the proportion of foods consumed per female, for each group separately, and used this table to calculate group-specific, diet-based Bray–Curtis dissimilarity matrices. For spatial proximity, we calculated the percent of time all adult female dyads spent within 5 m of each other during the same time period. Specifically, during each focal animal sample, the nearest adult female neighbor within 5 m is recorded at each 1-min point sample (893 points in Mica's group; 1637 points in Viola's group; range = 0–64 points per dyad). The proximity score between each pair of females (within groups) was calculated as the total number of point samples in which they were each other's nearest neighbors divided by the total number of point samples collected for each member of the dyad.

## Identification of socially structured taxa

To identify differentially abundant bacterial taxa by social group membership, we used the linear mixed model approach implemented in the program GEMMA (*Zhou and Stephens, 2012*), which allowed us to account for potential kinship effects in our data set. This approach assumes that the response variable (taxon abundance) is continuously distributed. To meet this assumption, we used methods established for analyzing high-throughput functional genomic data sets (*Rapaport et al., 2013*). Specifically, we first quantile normalized abundance values across individuals, focusing only on the 327 most prevalent taxa (i.e., those found in at least 50% of hosts based on our MetaPhlAn 2.0 analysis, regardless of abundance), and then transformed the distribution of values for each species to a standard normal. We then fit the following linear mixed model to the data for each species:

$$y = \mu + x\beta_x + a\beta_a + s\beta_s + r\beta_r + u + \varepsilon,$$

$$u \sim MVN(0, \sigma_u^2 K),$$

$$\varepsilon \sim MVN(0, \sigma_e^2 I).$$

Here, $y$ is the $n$ by 1 vector of normalized taxon abundances for the $n$ individuals in the sample; $\mu$ is the intercept; $x$ is the $n$ by 1 vector denoting social group membership; and $\beta_x$ is the effect size of social group membership. For the other covariates, $a$ is the $n$ by 1 vector denoting age and $\beta_a$ describes its effects on taxon abundance; $s$ is the $n$ by 1 vector denoting sex and $\beta_s$ its effect size; and $r$ is the $n$ by 1 vector denoting read depth and $\beta_r$ its effect size. The $n$ by 1 vector of $u$ is a random effects term to control for relatedness, and the $n$ by $n$ matrix K provides pedigree-based estimates of relatedness. Residual errors are represented by $\varepsilon$, an $n$ by 1 vector, and MVN denotes the multivariate normal distribution. We interpreted significantly non-zero $\beta_x$ values as support for differences in taxon abundance between social groups, using a false discovery rate threshold of 10% (*Storey and Tibshirani, 2003*) after checking that an empirically derived null distribution of p-values for this analysis was uniform (*Figure 3—figure supplement 1*).

To identify socially structured bacterial taxa within baboon social groups, we utilized a test of spatial autocorrelation, Moran's I, as implemented in the function *Moran.I* in the R package *ape* (*Paradis et al., 2004*). This analysis tests whether individuals with closer social bonds (as measured by the pairwise matrix of grooming strengths) tend to have more similar values for taxon abundance than those with weak or absent social bonds. Here, we again investigated the 327 most prevalent species from the MetaPhlAn 2.0 analysis. For this analysis, our power was constrained by the number of individuals in the social group. Thus, while we identified a large number of socially structured species within Viola's group (n = 51 of 327 species tested, at a false discovery rate of 10%), we did not observe strong evidence for socially structured species within Mica's group. Further investigation suggests this result is a consequence of sample size, as subsampling Viola's group (n = 29 individuals) to the size of Mica's group (n = 19 individuals) also resulted in little power to detect socially structured species. More than half of the time (58% of 100 random subsamples), fewer than 5 such cases were detected in

Viola's group after subsampling, and more than a third of the time (35%) no cases could be detected with the smaller sample size. Hence, we focused on results from Viola's group. We again used a 10% FDR threshold to identify significant taxa in this analysis, after ensuring that the empirical null distribution was uniform (*Figure 3—figure supplement 2*).

For both between-group and within-group analyses, we investigated enrichment of socially structured species in taxonomic units above the level of species (i.e., phylum, class, order, family, and genus) using hypergeometric tests. We required that taxonomic units include at least four species in our analysis to test for significant enrichment, and again employed an FDR threshold of 10%.

## Bacterial life style analysis

Descriptive data on bacteria were retrieved from the Genomes OnLine Database (GOLD; *Pagani et al., 2012*). This information included records for 34,533 unique entries and was downloaded from the GOLD website using a custom script on 02 June 2014 (available on GitHub at https://github.com/jklynch/scrape). Each record included fields for oxygen requirements and sporulation, as well as taxonomic classifications from the kingdom to species levels. We retained only completely sequenced genomes, and filtered this set to the entry, for any given species, associated with the most information about bacterial lifestyle and phenotype (n = 3818 unique species in 1280 unique genera). To assign 'genus-level' traits, we kept only genera in which all species in our filtered database were associated with the same trait value, if assigned (e.g., we assigned an anaerobic lifestyle to a genus only when all members of the genus were consistently anaerobic).

To investigate properties of significantly socially structured species, we merged the set of 327 prevalent species with the set of species with known lifestyle information. 138 species were represented in both sets; the comparable analysis at the genus level yielded n = 299 genera in both sets. We then applied hypergeometric tests to these data sets to ask whether socially structured species or genera, either between or within groups, were enriched for anaerobic, non-spore forming life-styles. Our results were broadly robust to whether anaerobes are distinguished in contrast to aerobes or in contrast to both aerobes and facultatively oxygen tolerant species (socially structured species between groups, species level traits: p = 0.025; socially structured species within group, species level traits: p = 0.100; socially structured species between groups, genus level traits: p = 0.056; socially structured species within group, genus level traits: p = 0.050).

## Alternative assessment of microbiome taxonomic composition using de novo assembled contigs

As an alternative to taxonomic profiling using MetaPhlAn 2.0, we also performed de novo contig assembly using the complete set of 1.4 billion raw reads. This approach allowed us to evaluate whether our results were robust to our methods for estimating species abundance. Reads were assembled de novo using Ray Meta, a short read de Bruijn assembler specifically devised for metagenome data, following the authors' recommendations (*Boisvert et al., 2012*). Bacterial proportions for each sample were then estimated using Ray Communities, utilizing all bacterial genomes available in GenBank and the Greengenes taxonomy as a reference (*DeSantis et al., 2006*). Summary statistics for alpha diversity and bacterial abundances estimated for each sample from the de novo assemblies can be found in *Supplementary files 3, 4*.

Across all 48 samples, we identified 1465 taxa that could be identified to the species level. Similar to our results using MetaPhlAn 2.0, we identified substantial representation of phyla typically found in gut microbiomes, including Bacteroidetes, Firmicutes, Proteobacteria, and Actinobacteria (*Figure 1—figure supplement 1*). The de novo assembly, however, identified a very large contribution of the phylum Spirochaetes in Viola's group (mean = 23.7%), which was primarily driven by the abundance of reads mapping to the bacteria *Treponema succinifaciens*. Notably, we also identified *T. succinifaciens* as significantly more abundant in Viola's group members than in Mica's group members using the MetaPhlAn approach (p = $2.46 \times 10^{-10}$). Thus, while our two approaches differed in the magnitude of this effect, the overall pattern was highly consistent.

The relationship between social group membership and gut microbiome composition using the de novo assembly approach broadly recapitulated the results using MetaPhlAn-based estimates (*Figure 2—figure supplement 1*). Specifically, social group membership explained 32.8% of global variation in gut microbial taxonomic composition, as summarized by a pairwise Bray–Curtis dissimilarity matrix (PERMANOVA: p < $1.0 \times 10^{-4}$). Kinship did not explain this relationship (partial Mantel test relating

group co-residency to taxonomic composition, controlling for pedigree-based kinship: r = 0.434, p < 1.0 × $10^{-5}$). Additionally, the de novo assembly approach again revealed that, within groups, closer grooming partners harbored more similar gut microbes (Mica's group: Mantel test $r$ = −0.197, p = 0.016; Viola's group: $r$ = −0.147, p = 1.9 × $10^{-3}$). However, while this relationship survives correcting for kinship in Viola's group ($r$ = −0.112, p = 0.017), it is not statistically detectable after controlling for kinship in Mica's group ($r$ = −0.091, p = 0.20). This pattern recapitulates our observations in the MetaPhlAn analysis, in which within group structuring of the microbiome tended to be weaker in Mica's group as well.

We next restricted the within-group grooming network analysis to adult females only, in order to test for alternative explanations for the grooming-microbiome composition effect. Grooming interactions remained a significant predictor of microbiome composition after accounting for both within-group patterns of dietary similarity (partial Mantel controlling for dietary similarity: Mica's group p = 0.038; Viola's: p = 0.006) and spatial proximity in Viola's group (partial Mantel controlling for proximity: Viola's: p = 0.009). In Mica's group, controlling for proximity produced a consistent trend with our main analyses, but eliminated the strong statistical signal of grooming on microbiome composition (Mica's group p = 0.124). We surmise that patterns of proximity, kinship, and grooming may be too closely correlated in Mica's group to disentangle in the de novo assembly-based data set, which may produce noisier estimates of taxonomic abundance.

## Data accessibility

Raw metagenomic sequencing data are deposited in NCBI's Short Read Archive (BioProject PRJNA271618). The custom script to download data from the Genomes Online Database (GOLD; *Pagani et al., 2012*) is available on GitHub at https://github.com/jklynch/scrape. Taxonomic and genic abundance data with sample metadata available on Dryad at doi:10.5061/dryad.8gp03 (*Tung et al., 2015*) and are uploaded as supplementary files associated with this manuscript.

## Acknowledgements

Support for the Amboseli Baboon Research Project was provided by the National Science Foundation (most recently IOS 1053461 to EAA and DEB 0919200 to SCA) and the National Institute of Aging (R01 AG034513 and P01 AG031719 to SCA and JA). We thank the Kenya Wildlife Services, Institute of Primate Research, National Museums of Kenya, National Commission for Science Technology and Innovation, members of the Amboseli-Longido pastoralist communities, Tortilis Camp, and Ker & Downey Safaris for their assistance in Kenya. Particular thanks to RS Mututua, S Sayialel, and JK Warutere, V Somen, and T Wango in Kenya, and K Pinc, N Learn, P Onyango and J Gordon in the US. Thanks also to S Morrow for her expertise in the lab, L David for helpful comments on earlier versions of this work, and Eric Alm, Jack Gilbert, and two anonymous reviewers for a careful review of an earlier version of this work. This work was carried out in part using computing resources at the Minnesota Supercomputing Institute.

## Additional information

### Funding

| Funder | Grant reference | Author |
| --- | --- | --- |
| National Science Foundation (NSF) | IOS 1053461 | Elizabeth A Archie |
| National Science Foundation (NSF) | DEB 0919200 | Susan C Alberts |
| National Institute on Aging | R01 AG034513 | Jeanne Altmann, Susan C Alberts |
| National Institute on Aging | P01 AG031719 | Jeanne Altmann, Susan C Alberts |

The funders had no role in study design, data collection and interpretation, or the decision to submit the work for publication.

### Author contributions

JT, LBB, RB, Conception and design, Analysis and interpretation of data, Drafting or revising the article; MBB, Analysis and interpretation of data, Drafting or revising the article; J-CG, JL, Analysis

and interpretation of data; LEG, Acquisition of data, Analysis and interpretation of data; JA, SCA, Acquisition of data; EAA, Conception and design, Acquisition of data, Analysis and interpretation of data, Drafting or revising the article

## Additional files

### Supplementary files

• Supplementary file 1.  Table listing all subject and sample characteristics.

• Supplementary file 2.  Table listing the relative abundance of microbial species in each sample inferred via MetaPhlAn 2.0.

• Supplementary file 3.  Table listing species richness and alpha diversity for each sample based on taxonomic profiling using MetaPhlAn 2.0 and *de novo* contig assembly.

• Supplementary file 4.  Table listing species proportional abundance for each sample based on *de novo* contig assembly.

• Supplementary file 5.  Table listing the relative abundance of enzyme gene orthologs in each sample.

• Supplementary file 6.  Table listing the proportional representation of common phyla in each sample. Taxonomic abundances were inferred for all samples using MetaPhlAn 2.0.

• Supplementary file 7.  Table listing the dietary composition during the microbiome sample collection period.

• Supplementary file 8.  Table listing statistical evidence for social structuring for the 327 most common bacterial species (prevalence >50% across 48 samples). Between group analyses were based on the linear mixed modeling approach implemented in the program GEMMA; q-values reflect a false discovery rate of 10%. Within-network analyses were based on Moran's I, as implemented in the function *Moran.I* in the R package *ape*; q-values reflect a false discovery rate of 10%.

### Major dataset

The following datasets were generated:

| Author(s) | Year | Dataset title | Dataset ID and/or URL | Database, license, and accessibility information |
| --- | --- | --- | --- | --- |
| Tung J, Barriero LB, Blekhman R, Archie EA | 2015 | Raw metagenomic sequencing data | http://www.ncbi.nlm.nih.gov/sra/?term=PRJNA271618 | Publicly available at the NCBI Short Read Archive (PRJNA271618). |
| Tung J, Barriero LB, Blekhman R, Archie EA | 2015 | Data from: Sample information and tables with microbial and genic relative abundance data | 10.5061/dryad.8gp03 | Available at Dryad Digital Repository under a CC0 Public Domain Dedication. |

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
