## [Decision Letter]

Thank you for sending your work entitled ‘Social networks predict gut microbiome composition in wild baboons’ for consideration at *eLife*. Your article has been favorably evaluated by Detlef Weigel (Senior editor), a guest Reviewing editor, and 3 reviewers.

The following individuals responsible for the peer review of your submission have agreed to reveal their identity: Eric Alm (Reviewing editor); Jack Gilbert (peer reviewer). Two other reviewers remain anonymous.

The Reviewing editor and the other reviewers discussed their comments before we reached this decision, and the Reviewing editor has assembled the following comments to help you prepare a revised submission.

We are pleased to report that the reviewers were enthusiastic about the novelty and impact of your findings, and were generally pleased with the rigor of your analyses. There was some discussion on whether metagenomic assembly might yield more direct evidence for transfer of specific strains of bacteria across social networks, but it was decided that the conclusions in the current manuscript were well supported, and that strain level analysis might be better suited to a follow up study.

However, it was felt that the statistical analysis leading to the identification of socially structured species could be made more rigorous. The main issue is that the procedure used to transform the data to fit a normal distribution could lead to slight inaccuracies in the p-values calculated by GEMMA. These inaccuracies could be further amplified by use of the FDR. To account for these effects, it was suggested that the procedure be empirically validated by shuffling the social group membership data, and verifying that no significant species are identified. It was also suggested that a similar empirical validation could add support to the within-group social network results. Finally, there were several areas in the text where p-values are given, but it is unclear what test was being run. These should be fully explained.

To summarize, please consider the following revisions:

1) Empirical validation of the procedure used to identify between social group socially structured species.

2) Empirical validation of the procedure used to identify within social group socially structured species.

3) Clarification of the statistical test used for all p-values reported.

---

## [Author Response]

*To summarize, please consider the following revisions*:

*1) Empirical validation of the procedure used to identify between social group socially structured species*.

Thank you for this suggestion. We have added an empirical validation of our modeling approach to identify socially structured species between social groups. Specifically, to confirm that our analysis did not bias us towards detecting false positives, we compared the signal in our true data set against an empirically derived null, obtained by permuting group membership across species abundances while keeping the modeling approach, kinship structure, and all other covariates constant. No differentially abundant species were identified in any of 10 permutations at a 10% FDR (the threshold used in our main analyses), and we obtained a uniform distribution of p-values for the empirical null. The difference between the null distribution and the true data set, which is highly enriched for a signature of social group, is shown using both histogram and quantile–quantile plots in the new Figure 3—figure supplement 1.

2) Empirical validation of the procedure used to identify within social group socially structured species.

We have added an empirical validation of our modeling approach to identify socially structured species within social groups. As above, we compared the signal in our true data set against an empirically derived null, this time permuting species abundance 10 times across group members while keeping the modeling approach and social network structure constant. No socially structured species (within group) were identified when analyzing permuted data using a 10% FDR, and we obtained a uniform distribution of p-values for the empirical null. We show supporting histogram and Q–Q plot figures comparing the true p-value distribution against p-values from the empirical null in the new Figure 3—figure supplement 2.

*3) Clarification of the statistical test used for all* p*-values reported*.

We have now clarified the statistical tests used for all reported p-values.